# Leveraging Importance Weights in Subset Selection

**Gui Citovsky**[1]**, Giulia DeSalvo**[1]**, Sanjiv Kumar**[1]**, Srikumar Ramalingam**[1]**,**
**Afshin Rostamizadeh**[1]**, Yunjuan Wang**[2]*

[1]Google Research, New York, NY, 10011
[2]Department of Computer Science, Johns Hopkins University, Baltimore, MD, 21218
`{gcitovsky,giuliad,sanjivk,rostami,rsrikumar}@google.com`
`ywang509@jhu.edu`

## Abstract

We present a subset selection algorithm designed to work with arbitrary model families in a practical batch setting. In such a setting, an algorithm can sample examples one at a time but, in order to limit overhead costs, is only able to update its state (i.e. further train model weights) once a large enough batch of examples is selected. Our algorithm, IWeS, selects examples by importance sampling where the sampling probability assigned to each example is based on the entropy of models trained on previously selected batches. IWeS admits significant performance improvement compared to other subset selection algorithms for seven publicly available datasets. Additionally, it is competitive in an active learning setting, where the label information is not available at selection time. We also provide an initial theoretical analysis to support our importance weighting approach, proving generalization and sampling rate bounds.

## 1 Introduction

Deep neural networks have shown remarkable success in several domains such as computer vision and natural language processing. In many tasks, this is achieved by heavily relying on extremely large labeled datasets. In addition to the storage costs and potential security/privacy concerns that come along with large datasets, training modern deep neural networks on such datasets also incur high computational costs. With the growing size of datasets in various domains, algorithm scalability is a real and imminent challenge that needs to be addressed. One promising way to solve this problem is with data subset selection, where the learner aims to find the most informative subset from a large number of training samples to approximate (or even improve upon) training with the entire training set. Such ideas have been extensively studied in k-means and k-median clustering (Har-Peled & Mazumdar, 2004), subspace approximation (Feldman et al., 2010), computational geometry (Agarwal et al., 2005), density estimation (Turner et al., 2021), to name a few.

One particular approach for solving data subsampling involves the computation of coresets, which are weighted subsets of a dataset that can act as the proxy for the whole dataset to solve some optimization task. Coreset algorithms are primarily motivated with theoretical guarantees that bound the difference between the training loss (or other such objective) over the coreset and that over the full dataset under different assumptions on the losses and hypothesis classes (Mai et al., 2021; Munteanu et al., 2018; Curtin et al., 2019; Karnin & Liberty, 2019). However, in practice, most competitive subset selection algorithms, that are designed for general loss functions and arbitrary function classes, focus only on selecting informative subsets of the data and typically do not assign weights to the selected examples. These methods are, for example, based on some notion of model uncertainty (Scheffer et al., 2001), information gain (Argamon-Engelson & Dagan, 1999), loss gradients (Paul et al., 2021; Ash et al., 2019), or diversity (Sener & Savarese, 2018). Counter to this trend, we show that weighting the selected samples can be very beneficial.

In this work, we present a subset selection algorithm called IWeS that is designed for general loss functions and hypothesis classes and that selects examples by importance sampling, a theoretically

---

*This work was done when the author was interning at Google.

motivated and unbiased sampling technique. Importance sampling is conducted according to a specially crafted probability distribution and, importantly, each sampled example is weighted inversely proportional to its sampling probability when computing the training loss. We develop two types of sampling probability for different practical requirements (e.g. computational constraints and label availability), but in both cases, the sampling probability is based on the example's entropy-based score computed using a previously trained model. We note, the IWeS algorithm is similar to the IWAL active learning algorithm of Beygelzimer et al. (2009) as both are based on importance sampling. However, in contrast to IWAL, IWeS uses a different sampling probability definition with a focus on providing a practical method that is amenable to large deep networks and complex hypothesis classes.

Through extensive experiments, we find that the IWeS algorithm is competitive for deep neural networks over several datasets. We compare our algorithm against four types of baselines whose sampling strategies leverage: the model's uncertainty over examples, diversity of selected examples, gradient information, and random sampling. Finally, we analyze a closely related albeit less practical algorithm that inspires the design of IWeS, called IWeS-V, proving it admits generalization and sampling rate guarantees that hold for general loss functions and hypothesis classes.

The contributions of this work can be summarized as follows:

1. We present the **I**mportance **We**ighted **S**ubset Selection (IWeS) algorithm that selects examples by importance sampling with a sampling probability based on a model's entropy, which is applicable to (and practical for) arbitrary model families including modern deep networks. In addition to the subset selection framework, IWeS also works in the active learning setting where the examples are unlabeled at selection time.

2. We demonstrate that IWeS achieves significant improvement over several baselines (Random, Margin, Least-Confident, Entropy, Coreset, BADGE) using VGG16 model for six common multi-class datasets (CIFAR10, CIFAR10-corrupted, CIFAR100, SVHN, Eurosat, Fashion MNIST), and using ResNet101 model for the large-scale multi-label OpenImages dataset.

3. We provide a theoretical analysis for a closely related algorithm, IWeS-V, in Section 4. We prove a $\mathcal{O}(1/\sqrt{T})$ generalization bound, which depends on the full training dataset size $T$. We further give a new definition of disagreement coefficient and prove a sampling rate bound by leveraging label information, which is tighter compared with the label complexity bound provided by Beygelzimer et al. (2009) that does not use label information.

## 1.1 RELATED WORK

**Uncertainty.** Uncertainty sampling, which selects examples that the model is least confident on, is favored by practitioners (Mussmann & Liang, 2018) and rather competitive among many recent algorithms (Yang & Loog, 2018). Uncertainty can be measured through entropy (Argamon-Engelson & Dagan, 1999), least confidence (Culotta & McCallum, 2005), and most popular is the margin between the most likely and the second most likely labels (Scheffer et al., 2001). More recent works measure the model uncertainty indirectly, such as selecting examples based on an estimated loss (Yoo & Kweon, 2019) as well as leveraging variational autoencoders and adversarial networks to find points not well represented by the current labeled data (Sinha et al., 2019; Kim et al., 2021). Beygelzimer et al. (2009) makes use of a disagreement-based notion of uncertainty and constructs an importance weighted predictor with theoretical guarantees called IWAL, which is further enhanced by Cortes et al. (2019). However, IWAL is not directly suitable for use with complex hypothesis spaces, such as deep networks, since it requires solving a non-trivial optimization over a subset of the hypothesis class, the so-called version space, in order to compute sampling probabilities. We further discuss these difficulties in Section 4.

**Diversity.** In another line of research, subsets are selected by enforcing diversity such as in the FASS (Wei et al., 2015) and Coreset (Sener & Savarese, 2018) algorithms. Wei et al. (2015) introduces a submodular sampling objective that trades off between uncertainty and diversity by finding a diverse set of samples from amongst those that the current trained model is most uncertain about. It was further explored by Kaushal et al. (2019) who designed a unified framework for data subset selection with facility location and dispersion-based diversity functions. Sener & Savarese (2018) show that the task of identifying a coreset in an active learning setting can be mapped to solving the k-center problem. Further recent works related to coreset idea are Mirzasoleiman et al. (2020); Killamsetty

et al. (2021), where the algorithms select representative subsets of the training data to minimize the estimation error between the weighted gradient of selected subset and the full gradient.

**Loss Gradient.** Another class of algorithms selects a subset by leveraging the loss gradients. For example, the GRAND score (Paul et al., 2021), or closely related EL2N score, leverages the average gradient across several different independent models to measure the importance of each sample. However, as such, it requires training several neural networks, which is computationally expensive. BADGE (Ash et al., 2019) is a sampling strategy for deep neural networks that uses k-MEANS++ on the gradient embedding of the networks to balance between uncertainty and diversity. ISAL (Liu et al., 2021) measures the potential impact of each example by leveraging the loss gradient and Hessian. Finally, for the sake of completeness, we note that importance weighting type approaches have also been used for the selection of examples within an SGD minibatch (Katharopoulos & Fleuret, 2018; Johnson & Guestrin, 2018), which can be thought of a change to the training procedure itself. In contrast, the problem setting we consider in this work requires explicitly producing a (weighted) subset of the training data and treats the training procedure itself as a black-box.

These are a sampling of data subset selection algorithms, and we refer the reader to (Guo et al., 2022) for a more detailed survey. In this work, we choose at least one algorithm from each of the categories mentioned above, in particular, Margin (Scheffer et al., 2001), BADGE (Ash et al., 2019), and Coreset (Sener & Savarese, 2018) to compare against empirically in Section 3. However, before that, we first formally define the IWeS algorithm in the following section.

## 2 THE IWES ALGORITHM

---

**Algorithm 1** Importance **We**ighted **S**ubset Selection (IWeS)

---

**Require:** Labeled pool $\mathcal{P}$, seed set size $k_0$, subset batch size $k$, number of iterations $R$, weight cap parameter $u$.
1: Initialize the subset $\mathcal{S} = \emptyset$.
2: $\mathcal{S}_0 \leftarrow$ Draw $k_0$ examples from $\mathcal{P}$ uniformly at random.
3: Set $\mathcal{S} = \{(x, y, 1) : (x, y) \in \mathcal{S}_0\}$ and $\mathcal{P} = \mathcal{P} \backslash \mathcal{S}_0$
4: **for** $r = 1, 2, \ldots, R$ **do**
5:     Set $\mathcal{S}_r = \emptyset$.
6:     Train $f_r, g_r$ on $\mathcal{S}$ using the weighted loss and independent random initializations.
7:     **while** $|\mathcal{S}_r| < k$ **do**
8:         Select $(x, y)$ uniformly at random from $\mathcal{P}$.
9:         Set $p(x, y)$ using entropy-based disagreement or entropy criteria shown in Eq (1) and (2).
10:         $Q \sim \text{Bernoulli}(p(x, y))$.
11:         **if** $Q = 1$ **then**
12:             Set $\mathcal{S}_r = \mathcal{S}_r \cup \left\{ \left( x, y, \min \left( \frac{1}{p(x,y)}, u \right) \right) \right\}$ and $\mathcal{P} = \mathcal{P} \backslash \{(x, y)\}$.
13:         **end if**
14:     **end while**
15:     $\mathcal{S} = \mathcal{S} \cup \mathcal{S}_r$.
16: **end for**
17: Train $f_{R+1}$ on $\mathcal{S}$ using the weighted loss.
18: **return** $\mathcal{S}, f_{R+1}$

---

We consider a practical batch streaming setting, where an algorithm processes one example at a time without updating its state until a batch of examples is selected. That is, like in standard streaming settings, the algorithm receives a labeled example, and decides whether to include it in the selected subset or not. Yet, the algorithm is only allowed to update its state after a fixed batch of examples have been selected in order to limit the overhead costs (e.g. this typically can include retraining models and extracting gradients). Unlike the pool-based setting where the algorithm receives the entire labeled pool beforehand, a batch streaming setting can be more appropriate when facing a vast training data pool since the algorithm can only process a subset of the pool without iterating over the whole pool. Note that any batch streaming algorithm can also be used in a pool-based setting, by simply streaming through the pool in a uniformly random fashion. At a high level, the IWeS algorithm selects examples by importance sampling where the sampling probability is based on the entropy of models trained on previously selected data. We define two sampling probabilities

that allow us to trade-off between performance and the computational cost, as well as label-aware or an active learning setting leading to less label annotation costs. As we will subsequently see, these sampling definitions are both easy to use and work well in practice.

To define the algorithm in more detail, we let $\mathcal{X} \in \mathbb{R}^d$ and $\mathcal{Y} = \{1, \ldots, c\}$ denote the input space and the multi-class label space, respectively. We assume the data $(x, y)$ is drawn from an unknown joint distribution $\mathcal{D}$ on $\mathcal{X} \times \mathcal{Y}$. Let $\mathcal{H} = \{h : \mathcal{X} \to \mathcal{Z}\}$ be the hypothesis class consisting of functions mapping from $\mathcal{X}$ to some prediction space $\mathcal{Z} \subset \mathbb{R}^{\mathcal{Y}}$ and let $\ell : \mathcal{Z} \times \mathcal{Y} \to \mathbb{R}$ denote the loss.

The pseudocode of IWeS is shown in Algorithm 1. Initially, a seed set $\mathcal{S}_0$ ($|\mathcal{S}_0| = k_0$) is selected uniformly at random from the labeled pool $\mathcal{P}$. Then the algorithm proceeds in rounds $r \in [1, \ldots, R]$ and it consists of two main components: training and sampling. At the training step at round $r$, the model(s) are trained using the importance-weighted loss, namely $f_r = \arg\min_{h \in \mathcal{H}} \sum_{(x,y,w) \in \mathcal{S}} w \cdot \ell(h(x), y)$ on the subset $\mathcal{S}$, selected so far, in the previous $r - 1$ rounds. Depending on the sampling strategy, we may need to randomly initialize two models $f_r, g_r$ in which case they are trained independently on the same selected subset $\mathcal{S}$, but with different random initializations. At the sampling step at round $r$, the IWeS algorithm calculates a sampling probably for example $(x, y) \in \mathcal{S}$ based on one of the following definitions:

- **Entropy-based Disagreement.** We define the sampling probability based on the disagreement on two functions with respect to entropy restricted to the labeled example $(x, y)$. That is,

$$p(x, y) = |P_{f_r}(y|x) \log_2 P_{f_r}(y|x) - P_{g_r}(y|x) \log_2 P_{g_r}(y|x)| \tag{1}$$

where $P_{f_r}(y|x)$ is the probability of class $y$ with model $f_r$ given example x. If the two functions, $f_r, g_r$, disagree on the labeled example $(x, y)$, then $p(x, y)$ will be small and the example will be less likely to be selected. This definition is the closest to the IWeS-V algorithm analyzed in Section 4 and achieves the best performance when the computational cost of training two models is not an issue. In Appendix A, we show an efficient version of entropy-based disagreement that utilizes only one model and achieves similar performance.

- **Entropy.** We define the sampling probability by the normalized entropy of the model $f_r$ trained on past selected examples:

$$p(x, \cdot) = -\sum_{y' \in \mathcal{Y}} P_{f_r}(y'|x) \log_2 P_{f_r}(y'|x) / \log_2 |\mathcal{Y}|. \tag{2}$$

The sampling probability $p(x, \cdot)$ is high whenever the model class probability $P_{f_r}(y'|x)$ is close to $1/|\mathcal{Y}|$, which is when the model is not confident about its prediction as it effectively randomly selects a label from $\mathcal{Y}$. This definition does not use the label $y$ and thus it can be used in an active learning setting where the algorithm can only access the unlabeled examples. Another advantage is that it only requires training one model, thereby saving some computational cost.

We note that entropy-based sampling has been used in algorithms such as uncertainty sampling as discussed in the related works section, but using entropy to define importance weights has not been done in past literature.

Based on one of these definitions, the IWeS algorithm then decides whether to include the example into the selected subset $\mathcal{S}$ by flipping a coin $Q$ with chosen sampling probability $p(x, y)$. If the example is selected, the example's corresponding weight $w$ is set to $\frac{1}{p(x,y)}$, and the example is removed from the labeled pool $\mathcal{P} = \mathcal{P} \setminus \{(x, y)\}$. This process is repeated until $k$ examples have been selected. Below we use IWeS-dis as an abbreviation for IWeS algorithm with Entropy-based Disagreement sampling probability and IWeS-ent for IWeS algorithm with Entropy sampling probability.

The weighted loss used to train the model can be written as $\frac{1}{|\mathcal{P}|} \sum_{i \in \mathcal{P}} \frac{Q_i}{p(x_i, y_i)} \ell(f(x_i), y_i)$ and it is an unbiased estimator of the population risk $\mathbb{E}_{(x,y) \sim \mathcal{D}}[\ell(f(x), y)]$. Yet such estimator can have a large variance when the model is highly confident in its prediction, that is whenever $P_{f_r}(y|x)$ is large, then $p(x, y)$ is small. This may lead to training instability and one pragmatic approach to addressing this issue is by "clipping" the importance sampling weights (Ionides, 2008; Swaminathan & Joachims, 2015). Thus in our algorithm, we let $u$ be the upper bound on the weight of the selected example. Although this clipping strategy introduces an additional parameter, we find it is not too sensitive and, as mentioned in the empirical section, set it to a fixed constant throughout our evaluation.

The primary computational cost of the algorithm, at each sampling round $r$, comes from (1) training the model and (2) the inference cost associated with computing the sampling probabilities on a

constant fraction of the unlabeled pool of examples. As described in the next section, besides for random sampling, all compared sampling algorithms incur similar re-training and model scoring costs at each sampling round.

# 3 EMPIRICAL EVALUATION

We compare IWeS with state-of-the-art baselines on several image classification benchmarks. Specifically, we consider six multi-class datasets (CIFAR10, CIFAR100 (Krizhevsky & Hinton, 2009), SVHN (Netzer et al., 2011), EUROSAT (Helber et al., 2019), CIFAR10 Corrupted (Hendrycks & Dietterich, 2019), Fashion MNIST (Xiao et al., 2017) and one large-scale multi-label OpenImages dataset (Krasin et al., 2017). In the multi-class setting, each image is associated with only one label. On the other hand, the multi-label OpenImages dataset consists of 19,957 classes over 9M images, where each image contains binary labels for a small subset of the classes (on average 6 labels per image). Further details of each dataset can be found in Table 1 and Table 2 in the appendix.

For all experiments, we consider a diverse set of standard baselines from both subset selection and active learning literature (discussed in Section 1.1).

- **Uncertainty Sampling** selects top $k$ examples on which the current model admits the highest uncertainty. There are three popular ways of defining model uncertainty $s(\mathrm{x})$ of an example x, namely margin sampling, entropy sampling, and least confident sampling, and all are based on $\mathrm{P}_f[\hat{y}|\mathrm{x}]$, the probability of class $\hat{y}$ given example x according to the model $f$. Margin sampling defines the model uncertainty of an example x as $s(\mathrm{x}) = 1 - (\mathrm{P}_f[\hat{y}_1|\mathrm{x}] - \mathrm{P}_f[\hat{y}_2|\mathrm{x}])$ where $\hat{y}_1 = \mathrm{argmax}_{y\in\mathcal{Y}} \mathrm{P}_f[y|\mathrm{x}], \hat{y}_2 = \mathrm{argmax}_{y\in\mathcal{Y}\setminus y_1} \mathrm{P}_f[y|\mathrm{x}]$ are the first and second most probable classes for model $f$. For entropy sampling, model uncertainty is defined as $s(\mathrm{x}) = -\sum_{y\in\mathcal{Y}} \mathrm{P}_f(\hat{y}|\mathrm{x}) \log(\mathrm{P}_f(\hat{y}|\mathrm{x}))$ while for least confidence sampling, it is defined as $s(\mathrm{x}) = 1 - \max_{y\in\mathcal{Y}} \mathrm{P}_f(\hat{y}|\mathrm{x})$.

- **BADGE** of Ash et al. (2019) selects $k$ examples by using the $k$-MEANS++ seeding algorithm using the gradient vectors, computed with respect to the penultimate layer using the most likely labels given by the latest model checkpoint.

- **Coreset ($k$-Center)** of Sener & Savarese (2018) selects a subset of examples using their embeddings derived from the penultimate layer using the latest model checkpoint. In particular, the $k$ examples are chosen using a greedy 2-approximation algorithm for the $k$-center problem.

- **Random Sampling** selects $k$ examples uniformly at random.

## 3.1 MULTI-CLASS EXPERIMENTS

Here, we compare the IWeS algorithm against the baselines on the six multi-class image datasets. We use the VGG16 architecture with weights that were pre-trained using ImageNet as well as add two fully-connected 4096 dimensional layers and a final prediction layer. Xavier uniform initialization is used for the final layers. For each dataset, we tune the learning rate by choosing the rate from the set $\{0.001, 0.002, 0.005, 0.01, 0.1\}$ that achieves best model performance on the seed set. We use batch SGD with the selected learning rate and fix SGD's batch size to 100. At each sampling round $r$, the model is trained to convergence on all past selected examples for at least 20 epochs. For IWeS, we set the weight capping parameter to 2 for all datasets except for CIFAR10 which we decreased to 1.5 in order to reduce training instability.

The embedding layer for BADGE and Coreset is extracted from the penultimate layer having a dimension of 4096. The effective dimension of the gradient vector in BADGE grows with the number of labels, which is problematic for CIFAR100 as it has 100 classes. More specifically, the runtime of BADGE is given by $\mathcal{O}(dkT)$, which can be large for CIFAR100 since the dimension of the gradient vector from the penultimate layer is $d = 4096 \times 100$, the size of the labeled pool is $T$=50K, and the number of examples selected in each round is $k$=5K. In order to solve this inefficiency for CIFAR100, we split the labeled pool randomly into 100 partitions and ran separate instances of the algorithm in each partition with batch size $k/100$.

Each algorithm is initialized with a seed set that is sampled uniformly at random from the pool. After that, sampling then proceeds in a series of rounds $r$ where the model is frozen until a batch $k$ of examples is selected. The seed set size and sampling batch size $k$ are set to 1K for CIFAR10, SVHN,

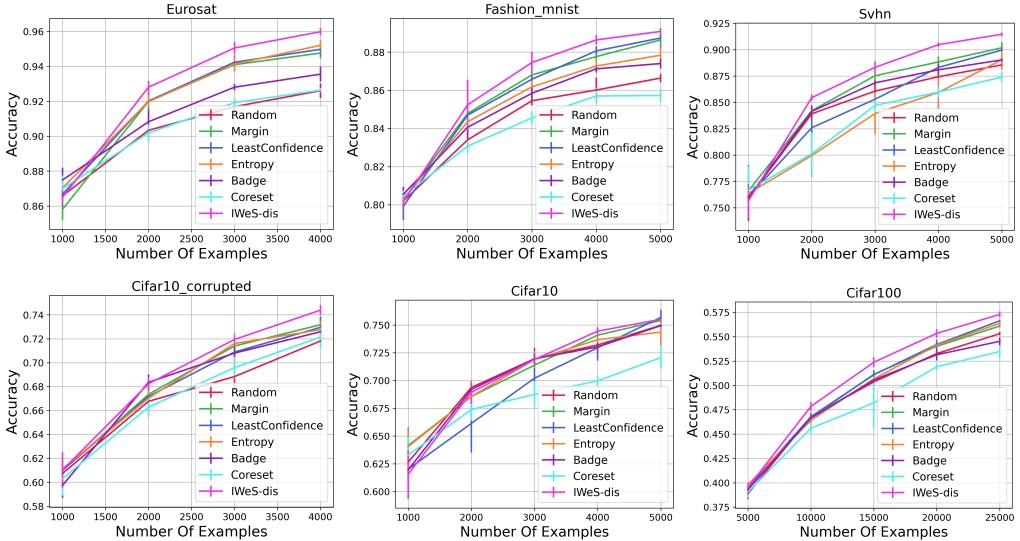

Figure 1: Accuracy of VGG16 when trained on examples selected by IWeS-dis and baseline algorithms.

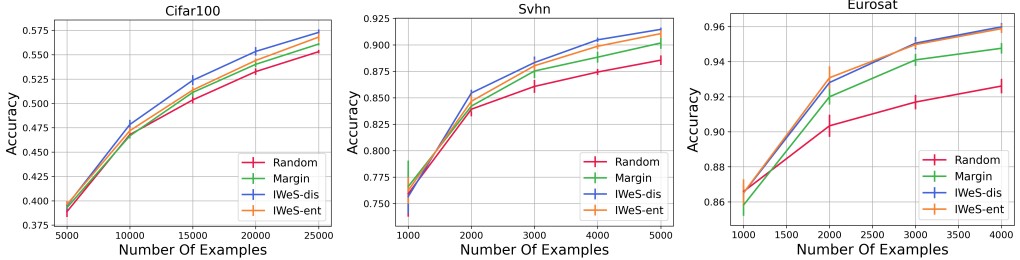

Figure 2: Accuracy of VGG16 when trained on examples selected by IWeS-ent, IWeS-dis, margin sampling and random sampling.

EUROSAT, CIFAR10 Corrupted, Fashion MNIST, and to 5K for CIFAR100. The initial seed set size allows at least 50 initial examples per class on average. The experiment was repeated for 5 trials. Any trial that encountered divergent training, i.e. the resulting model accuracy is more than three times below the standard error of model's accuracy on seed set, was dropped. We note that this happened infrequently (less than 3% of the time) and all reported averaged results have at least 3 trials.

Figure 1 shows the mean and standard error of VGG16 model's accuracy on a held out test set comparing IWeS-dis to the baseline methods. The IWeS-dis algorithm either outperforms or matches the performance of the baseline algorithms for all datasets. We also find that margin sampling consistently performs well against the remaining baseline algorithms and that BADGE either matches the performance of margin sampling or slightly underperforms on some datasets (Eurosat, Fashion MNIST, CIFAR100). Coreset admits a similar and at times slightly poorer performance compared to random sampling.

Next, Figure 2 compares the two variants of our algorithm: IWeS-dis and IWeS-ent. We find that the IWeS-dis performs slightly better than IWeS-ent on most of the datasets. This is not surprising since the IWeS-dis sampling probability leverages label information and more computational power, i.e. trains two models. As explained in Section 4, it also better fits our theoretical motivation. Nevertheless, it is important to note that IWeS-ent, without the label information, still consistently outperforms or matches the performance of the baselines for all the datasets.

### 3.2 MULTI-LABEL OPENIMAGES EXPERIMENTS

Here, we evaluate the performance of the IWeS algorithm on OpenImages v6. We train a ResNet101 model implemented using tf-slim on 64 Cloud two core TPU v4 acceleration, and apply batch SGD with batchsize of 6144 and an initial learning rate of $10^{-4}$ with decay logarithmically every $5 \times 10^8$

examples. We add a global pooling layer with a fully connected layer of 128 dimensions as the final layers of the networks, which is needed by BADGE and Coreset. The model is initialized with weights that were pre-trained on the validation split using 150K SGD steps, and at each sampling round, the model is trained on all past selected examples with an additional 15K SGD steps.

In the previous section, our results show that the IWeS-dis algorithm only slightly outperforms the IWeS-ent algorithm on a few datasets. Also, since the IWeS-dis requires training two neural networks, which is computationally expensive in this scenario, we only test the performance of IWeS-ent. Since IWeS-ent does not use the label information, we can also measure the performance of the algorithm in an active learning setting.

Since OpenImages is a multi-label dataset, the sampling algorithms must not only select the image, but also the class. That is, each example selected by an algorithm consists of an image-class pair with a corresponding binary label indicating whether the corresponding class is present in the image or not. In order to adapt IWeS-ent to the multi-label setting, the entropy

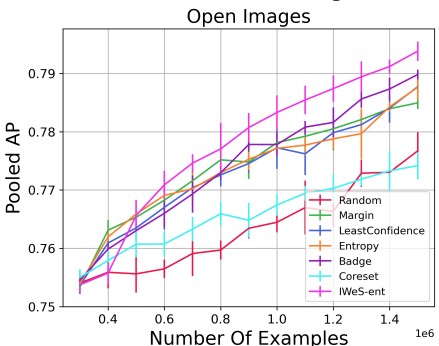

Figure 3: Pooled Average Precision of ResNet101 trained on examples selected by IWeS-ent and the baseline algorithms.

sampling probability for each image-class pair is defined as $p(\mathbf{x}, \cdot) = -\mathrm{P}_{f_r}(y|\mathbf{x}) \log_2 \mathrm{P}_{f_r}(y|\mathbf{x}) - (1 - \mathrm{P}_{f_r}(y|\mathbf{x})) \log_2 (1 - \mathrm{P}_{f_r}(y|\mathbf{x}))$, where $\mathrm{P}_{f_r}(y|\mathbf{x})$ is the model class probability of a positive label at round $r$. A seed set of size 300K is sampled uniformly at random from the pool, and at each sampling round $r$, the algorithms select 100K examples. Similarly to the previous section, in order to run BADGE on OpenImages, we divide the pool into 100 partitions and run separate instances of the algorithm in each partition. For IWeS, the weight capping parameter is set to 10.

Figure 3 shows the mean and standard error across 5 trials of the pooled average precision (Pooled AP) metric for each algorithm. As the number of selected examples increases, IWeS-ent outperforms all other baselines methods on the OpenImages dataset. We also find that BADGE performs similarly or even slightly worse than the uncertainty-based sampling algorithms when the number of selected examples is smaller than 800K, and then outperforms all uncertainty-based sampling as the number of selected examples increases. Coreset initially performs better than random sampling, but at later sampling rounds, it admits a similar performance to random sampling.

## 4  THEORETICAL MOTIVATION

In order to theoretically motivate the IWeS algorithm, we analyze a closely related algorithm which we call IWeS-V, adapted from the IWAL algorithm of Beygelzimer et al. (2009). We prove that IWeS-V admits generalization bounds that scale with the dataset size $T$ and sampling rate bounds that are in terms of a new disagreement coefficient tailored to the subset selection framework.

Below, we let $L(h) = \mathbb{E}_{(\mathbf{x},y) \sim \mathcal{D}}[\ell(h(\mathbf{x}), y)]$ denote the expected loss of hypothesis $h \in \mathcal{H}$, and $h^* = \operatorname{argmin}_{h \in \mathcal{H}} L(h)$ be the best-in-class hypothesis. Without loss of generality, we consider a bounded loss $\ell : \mathcal{Z} \times \mathcal{Y} \to [0, 1]$ mapping to the interval $[0, 1]$. Such a loss can be achieved by any bounded loss after normalization. For simplicity, we assume $\mathcal{H}$ is a finite set, but our results can be easily extended by standard covering arguments to more general hypothesis sets such as finite VC-classes.

The IWeS-V algorithm operates on an i.i.d. example $(\mathbf{x}_1, y_1), (\mathbf{x}_2, y_2), \ldots, (\mathbf{x}_T, y_T)$ drawn from $\mathcal{D}$ sequentially. It maintains a version space $\mathcal{H}_t$ at any time $t$, with $\mathcal{H}_1 = \mathcal{H}$. At time $t$, IWeS-V flips a coin $Q_t \in \{0, 1\}$ with bias $p_t$ defined as

$$p_t = \max_{f,g \in \mathcal{H}_t} \ell(f(\mathbf{x}_t), y_t) - \ell(g(\mathbf{x}_t), y_t) \tag{3}$$

where $\mathcal{H}_t = \left\{ h \in \mathcal{H}_{t-1} : \frac{1}{t} \sum_{s=1}^{t} \frac{Q_s}{p_s} \ell(h(\mathbf{x}_s), y_s) \le \min_{h' \in \mathcal{H}_{t-1}} \frac{1}{t} \sum_{s=1}^{t} \frac{Q_s}{p_s} \ell(h'(\mathbf{x}_s), y_s) + \Delta_{t-1} \right\}$ with $\Delta_{t-1} = \sqrt{\frac{8 \log(2T(T+1)|\mathcal{H}|^2/\delta)}{t-1}}$. The example is selected if $Q_t = 1$ and otherwise it is discarded. The main idea behind this algorithm is thus to define a sampling probability that is in terms of the disagreement between two hypothesis $f, g$ that are not too far from the best model trained on the past selected data, i.e. $\min_{h \in \mathcal{H}_{t-1}} \frac{1}{t} \sum_{s=1}^{t} \frac{Q_s}{p_s} \ell(h(\mathbf{x}_s), y_s)$. The formal IWeS-V algorithm pseudo-code (Algorithm 2) and all the theorem proofs can be found in Appendix B.

For general, e.g. non-linear, hypothesis classes it is computationally infeasible to find two hypotheses $f, g \in \mathcal{H}_t$ that maximize the expression in equation (3). This main impracticality of IWeS-V is reason why we developed the IWeS algorithm of the previous section. This drawback is also shared by the IWAL algorithm of Beygelzimer et al. (2009), which computes a sampling probability very similar to that of equation (3), but with an additional maximization over the choice of $y \in \mathcal{Y}$ in the definition of the sampling probability $p_t$.

Before continuing we explain how our practical algorithm IWeS-dis, specifically using sampling probability in equation (1), is closely related to the IWeS-V algorithm. Recall that the IWeS algorithm trains two models $f$ and $g$ each minimizing the importance-weighted loss using the data sampled so far. Therefore, each model exhibits reasonable training loss, i.e. they are expected to be included in the version space $\mathcal{H}_t$ of good hypothesis, while the different random initializations (in the case of non-convex neural network hypotheses) results in models that still differ in certain regions of the feature space. Thus, the difference in equation (1) can be thought of as a less aggressive version of the difference found in the maximization of equation (3).

Another dissimilarity between the two is that the IWeS-dis algorithm is defined for the batch streaming setting while the IWeS-dis algorithm and its analysis is developed for the streaming setting. Said differently, the IWeS-V algorithm can be seen as a special case of the IWeS-dis algorithm with target subset size of 1. To extend the theoretical guarantees of IWeS-V to the batch streaming setting, we can follow a similar analysis developed by Amin et al. (2020) to also find that the effects of delayed feedback in the batch streaming setting are in fact mild as compared to the streaming setting.

## 4.1 GENERALIZATION BOUND

Next, we turn to the topic of generalization guarantees and we review an existing bound for coreset based algorithms. The guarantees of coreset algorithms are generally focused on showing that a model's training loss on the selected subset is close to the same model's training loss on the whole dataset. That is, given dataset $\mathcal{P} = \{(x_i, y_i)\}_{i=1}^{T} \sim \mathcal{D}^T$, the learner seek to select a subset $m < T$ of examples $\mathcal{S} = \{(x_i', y_i')\}_{i=1}^{m}$ along with a corresponding set of weights $w_1, \ldots, w_m$ such that for some small $\epsilon > 0$ and for all $h \in \mathcal{H}$, the *additive error coreset guarantee* holds $\left| \sum_{i=1}^{m} w_i \ell(h(x_i'), y_i') - \sum_{i=1}^{T} \ell(h(x_i), y_i) \right| \leq \epsilon T$. The following proposition, which is a minor extension of Fact 8 of Karnin & Liberty (2019), allows us to convert a coreset guarantee into a generalization guarantee.

**Proposition 4.1.** Let $h' = \operatorname{argmin}_{h \in \mathcal{H}} \sum_{i=1}^{m} w_i \ell(h(x_i'), y_i')$, and let the additive error coreset guarantee hold for any $\epsilon > 0$, then for any $\delta > 0$, with probability at least $1 - \delta$, it holds that $L(h') \leq L(h^*) + 2\epsilon + 2\sqrt{\ln(4/\delta)/2T}$.

As shown above, the generalization guarantee depends linearly on $\epsilon$ which in turn depends on the size of the subset $m$. To give a few examples, Karnin & Liberty (2019) show that for hypotheses that are defined as analytic functions of dot products (e.g. generalized linear models) this dependence on $m$ is $\epsilon = O(1/m)$, while for more complex Kernel Density Estimator type models the dependence is $\epsilon = O(1/\sqrt{m})$. See Mai et al. (2021), Table 1, for examples on the dependency between $\epsilon$ and $m$ under different data distributions assumptions (e.g. uniform, deterministic, $\ell_1$ Lewis) and for specific loss functions (e.g. log loss, hinge loss).

We now provide a generalization guarantee for the IWeS-V algorithm, which depends on the size of the labeled pool size $T$. The proof follows from that in Beygelzimer et al. (2009).

**Theorem 4.2.** Let $h^* \in \mathcal{H}$ be the minimizer of the expected loss function $h^* = \operatorname{argmin}_{h \in \mathcal{H}} L(h)$. For any $\delta > 0$, with probability at least $1 - \delta$, for any $t \geq 1$ with $t \in \{1, 2 \ldots, T\}$, we have that $h^* \in \mathcal{H}_t$ and that $L(f) - L(g) \leq 2\Delta_{t-1}$ for any $f, g \in \mathcal{H}_t$. In particular, if $h_T$ is the output of IWeS-V, then $L(h_T) - L(h^*) \leq 2\Delta_{T-1} = \mathcal{O}\big(\sqrt{\log(T/\delta)/T}\big)$.

Unlike the distribution-specific and loss-specific theoretical guarantees proposed in the coreset literature, Theorem 4.2 holds for any bounded loss function and general hypothesis classes. If we ignore log terms and consider the more complex Kernel Density Estimator class of hypotheses, the coreset method of Karnin & Liberty (2019) requires $m = \mathcal{O}(T)$ coreset samples in order to achieve an overall $\mathcal{O}(1/\sqrt{T})$ generalization bound. As we will see in the next section, the required IWeS sampling rate can also be as high as $\mathcal{O}(T)$, but critically is scaled by the best-in-class loss, which in favorable cases is significantly smaller than one.

### 4.2 SAMPLING RATE BOUNDS

Hanneke (2007) proves that the expected number of labeled examples needed to train a model in an active learning setting can be characterized in terms of the disagreement coefficient of the learning problem. Later, Beygelzimer et al. (2009) generalizes this notion to arbitrary loss functions, and in this work, we further generalize this for the subset selection setting.

Recall that the disagreement coefficient $\theta_{\mathrm{AL}}$ in Beygelzimer et al. (2009) for the active learning setting is defined as

$$\theta_{\mathrm{AL}} = \sup_{r \geq 0} \frac{\mathbb{E}_{\mathrm{x} \sim \mathcal{X}} \left[ \max_{h \in \mathcal{B}_{\mathrm{AL}}(h^*, r)} \max_{y \in \mathcal{Y}} |\ell(h(x), y) - \ell(h^*(x), y)| \right]}{r},$$

where $\mathcal{B}_{\mathrm{AL}}(h^*, r) = \{h \in \mathcal{H} : \rho_{\mathrm{AL}}(h, h^*) \leq r\}$ with $\rho_{\mathrm{AL}}(f, g) = \mathbb{E}_{\mathrm{x} \sim \mathcal{X}}[\sup_{y \in \mathcal{Y}} |\ell(f(\mathrm{x}), y) - \ell(g(\mathrm{x}), y)|]$. Informally, this coefficient quantifies how much disagreement there is among a set of classifiers that is close to the best-in-class hypothesis. In the subset selection setting, labels are available at sample time and, thus, we are able to define the following disagreement coefficient:

**Definition 4.1.** Let $\rho_{\mathrm{S}}(f, g) = \mathbb{E}_{(\mathrm{x}, y) \in \mathcal{D}}[|\ell(f(\mathrm{x}), y) - \ell(g(\mathrm{x}), y)|]$ and $\mathcal{B}_{\mathrm{S}}(h^*, r) = \{h \in \mathcal{H} : \rho_{\mathrm{S}}(h, h^*) \leq r\}$ for $r \geq 0$. The disagreement coefficient in the subset selection setting is defined as

$$\theta_{\mathrm{S}} = \sup_{r \geq 0} \frac{\mathbb{E}_{(\mathrm{x}, y) \sim \mathcal{D}} \left[ \max_{h \in \mathcal{B}_{\mathrm{S}}(h^*, r)} |\ell(h(\mathrm{x}), y) - \ell(h^*(\mathrm{x}), y)| \right]}{r}.$$

The main difference between the above coefficient and that of Beygelzimer et al. (2009) is that there is no supremum over all label $y \in \mathcal{Y}$ both in the definition of the distance $\rho$ and the coefficient's numerator. Instead, the supremum is replaced with an expectation over the label space.

The following theorem leverages $\theta_{\mathrm{S}}$ to derive an upper bound on the expected number of selected examples for the IWeS-V algorithm. Below, let $\mathcal{F}_t = \{(\mathrm{x}_i, y_i, Q_i)\}_{i=1}^t$ be the observations of the algorithm up to time $t$.

**Theorem 4.3.** For any $\delta > 0$, with probability at least $1 - \delta$, the expected sampling rate of the IWeS-V algorithm is: $\sum_{t=1}^T \mathbb{E}_{(\mathrm{x}_t, y_t) \sim \mathcal{D}} \left[ p_t | \mathcal{F}_{t-1} \right] = \mathcal{O}\left( \theta_{\mathrm{S}} \left( L(h^*)T + \sqrt{T \log(T/\delta)} \right) \right)$.

Suppressing lower order terms, the above expected sampling rate bound is small whenever the product of the disagreement coefficient and the expected loss of the best-in-class is small. In such cases, by combining the above theorem with the generalization guarantee, it holds that IWeS-V returns a hypothesis trained on a only fraction of the points that generalizes as well as a hypothesis trained on the full dataset of size $T$. Theorem 4.3 can be further improved by adapting the ideas found in Cortes et al. (2019) to the IWeS-V algorithm. See Appendix B.4 for this enhanced analysis.

The form of this sampling rate bound is similar to that of Beygelzimer et al. (2009). More concretely, under the assumption that a loss function has bounded slope asymmetry, that is $K_\ell = \sup_{z, z' \in \mathcal{Z}} \frac{\max_{y \in \mathcal{Y}} |\ell(z, y) - \ell(z', y)|}{\min_{y \in \mathcal{Y}} |\ell(z, y) - \ell(z', y)|}$ is bounded, with probability at least $1 - \delta$, the expected number of examples selected by the IWAL algorithm is given by $\mathcal{O}\left( \theta_{\mathrm{AL}} K_\ell \left( L(h^*)T + \sqrt{T \log(T/\delta)} \right) \right)$.

Thus, the main difference between the sampling rate bound of the IWAL algorithm and the IWeS-V algorithm are the factors that depends on the two disagreement coefficients: $\theta_{\mathrm{AL}} K_\ell$ and $\theta_{\mathrm{S}}$. Since $\theta_{\mathrm{S}}$ leverages the label information we may expect it to provide a tighter bound, compared to using the label-independent disagreement $\theta_{\mathrm{AL}}$. Theorem 4.4 shows that this is indeed the case.

**Theorem 4.4.** If the loss function has a bounded slope asymmetry $K_\ell$, then $\theta_{\mathrm{S}} \leq \theta_{\mathrm{AL}} K_\ell$.

The above theorem in conjunction with the sampling rate guarantees thus proves that the sampling rate bound of IWeS of Theorem 4.3 is tighter than the sampling rate bound of the IWAL algorithm.

## 5 CONCLUSION

In this paper we have introduced a subset selection algorithm, IWeS that is designed for arbitrary hypothesis classes including deep networks. We have shown that the IWeS algorithm outperforms several natural and important baselines across multiple datasets. In addition, we have developed an initial theoretical motivation for our approach based on the importance weighted sampling mechanism. A natural next step is enforcing a notion of diversity as it will likely provide improved performance in the large-batch sampling setting and thus, we plan to adapt the diversity-based method in Citovsky et al. (2021) by replacing the uncertainty sampling component with the IWeS algorithm.

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

# Supplementary Material

## A    EXTENDED ALGORITHMIC AND EXPERIMENTAL DETAILS

### A.1    DATASET DETAILS

Table 1 and Table 2 provide further details of the six multi-class classification datasets and the multi-label Open Images dataset, respectively.

|  | Train | Test | # Classes | Image Size | Description |
|---|---|---|---|---|---|
| CIFAR10 | 50,000 | 10,000 | 10 | 32×32×3 | classify the object in the image |
| SVHN | 73,257 | 26,032 | 10 | 32×32×3 | classify street view house number |
| CIFAR Corrupted | 7,614 | 2,386 | 10 | 32×32×3 | classify the corrupted object in the image |
| Eurosat | 8,000 | 5,000 | 10 | 64×64×3 | classify land use and land cover satelite image |
| Fashion MNIST | 60,000 | 10,000 | 10 | 32×32×3 | classify the type of clothes |
| CIFAR100 | 50,000 | 10,000 | 100 | 32×32×3 | classify the object in the image |

Table 1: Multi-class Classification Datasets statistics

|  | Images | Positives | Negatives |
|---|---|---|---|
| Train | 9,011,219 | 19,856,086 | 37,668,266 |
| Validation | 41,620 | 367,263 | 228,076 |
| Test | 125,436 | 1,110,124 | 689,759 |

Table 2: Open Images Dataset v6 statistics by data split

### A.2    MORE EXPERIMENT DETAILS AND RESULTS

When running IWeS, at each iteration, we pass over the labeled pool sequentially in a uniform random order, removing each selected example from the pool. If we exhaust the entire pool before selecting $k$ examples, we start again at the beginning of the sequence and iterate over the remaining examples. In order to reduce the number of passes required for the smaller datasets, we scale the sampling probabilities of the remaining points uniformly by $1 + \frac{j}{10}$, where $j$ is the number of passes so far.

Figure 4 compares IWeS-dis and IWeS-ent for additional three datasets that we omit in Section 3.

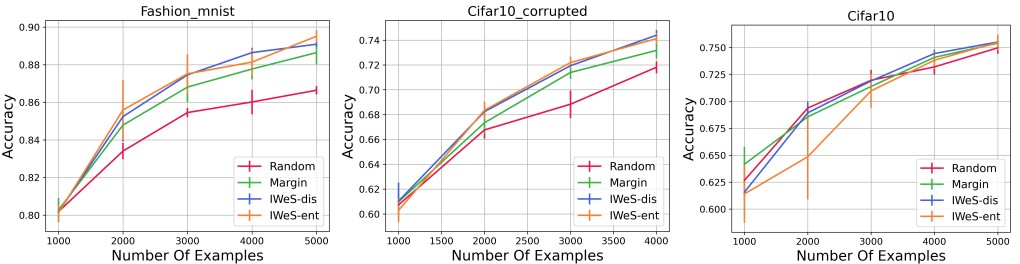

Figure 4: Accuracy of VGG16 trained on examples selected by IWeS-ent, IWeS-dis, margin sampling and random sampling.

### A.3    EFFICIENT VERSION OF ENTROPY-BASED DISAGREEMENT

Here we develop an efficient version of entropy-based disagreement which we call IWeS-loss by defining the sampling probability to be proportional to the model's entropy restricted to the labeled examples $(\mathbf{x}, y)$ as follows:

$$p(\mathbf{x}, y) = -\mathrm{P}_{f_r}(y|\mathbf{x}) \log \mathrm{P}_{f_r}(y|\mathbf{x}). \tag{4}$$

where $\mathbf{P}_{f_r}(y|\mathbf{x})$ is the probability of class $y$ with model $f_r$ given example x. As $\mathbf{P}_{f_r}(y|\mathbf{x})$ increase from 0 to 1, this sampling probability first increase then decrease. Thus the sampling probability is high whenever the model is not confident about the model prediction. Unlike IWeS-dis, this definition only requires training one model, thereby saving some computational cost.

Figure 5 compares the two variants IWeS-dis and IWeS-loss. We find that the performance are similar across all datasets, with IWeS-loss behave slightly better than IWeS-dis at the first two sample iterations (i.e. Fashion MNIST, CIFAR10, Eurosat), and IWeS-dis slightly outperform IWeS-loss as the number of sample iteration increases (i.e. SVHN, Eurosat).

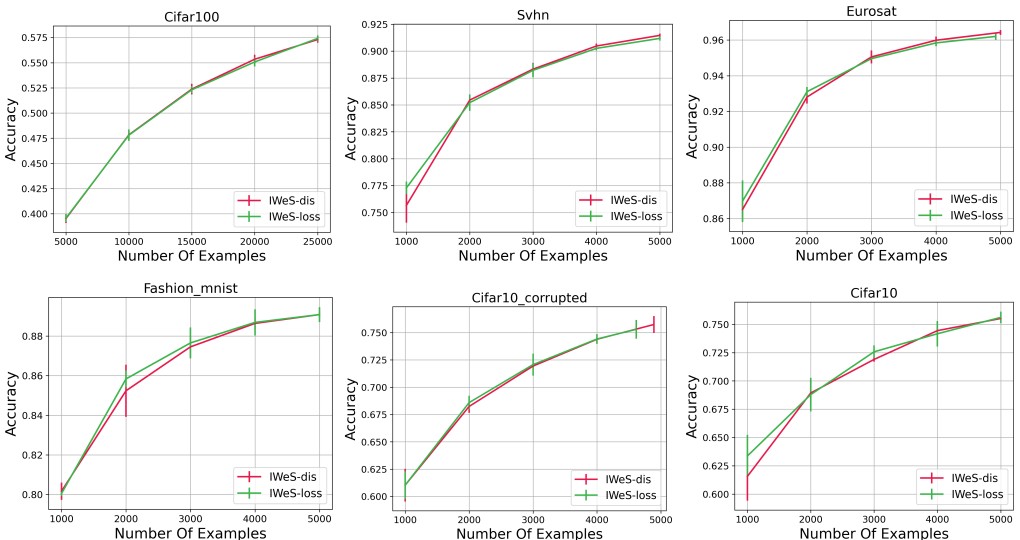

Figure 5: Accuracy of VGG16 trained on examples selected by IWeS-loss and IWeS-dis

# B    PROOFS OF THEORETICAL GUARANTEE

## B.1    PROOF OF PROPOSITION 4.1

**Proposition 4.1.** Let $h' = \operatorname{argmin}_{h \in \mathcal{H}} \sum_{i=1}^{m} w_i \ell(h(\mathbf{x}_i'), y_i')$, and let the additive error coreset guarantee hold for any $\epsilon > 0$, then for any $\delta > 0$, with probability at least $1 - \delta$, it holds that $L(h') \leq L(h^*) + 2\epsilon + 2\sqrt{\ln(4/\delta)/2T}$.

*Proof of Proposition 4.1.*

$$L(h') = \mathbb{E}_{(\mathbf{x},y)\sim\mathcal{D}}[\ell(h'(\mathbf{x}), y)]$$

$$\leq \frac{1}{T}\sum_{i=1}^{T} \ell(h'(\mathbf{x}_i), y_i) + \sqrt{\frac{\ln(4/\delta)}{2T}} \qquad \text{(Hoeffding inequality)}$$

$$\leq \frac{1}{T}\sum_{i=1}^{m} w_i \ell(h'(\mathbf{x}_i'), y_i') + \epsilon + \sqrt{\frac{\ln(4/\delta)}{2T}} \qquad (0 \leq \ell(\cdot) \leq 1)$$

$$\leq \frac{1}{T}\sum_{i=1}^{m} w_i \ell(h^*(\mathbf{x}_i'), y_i) + \epsilon + \sqrt{\frac{\ln(4/\delta)}{2T}} \qquad \text{(By the definition of } h')$$

$$\leq \frac{1}{T}\sum_{i=1}^{T} \ell(h^*(\mathbf{x}_i), y_i) + 2\epsilon + \sqrt{\frac{\ln(4/\delta)}{2T}}$$

$$\leq \mathbb{E}_{(\mathbf{x},y)\sim\mathcal{D}}[\ell(h^*(\mathbf{x}), y)] + 2\epsilon + 2\sqrt{\frac{\ln(4/\delta)}{2T}} \qquad \text{(Hoeffding inequality)}$$

$$= L(h^*) + 2\epsilon + 2\sqrt{\frac{\ln(4/\delta)}{2T}}$$

$\square$

## B.2 Proofs for statements in Section 4.1

Algorithm 2 contains a detailed pseudocode of the IWeS-V algorithm referred to in the main body of the paper.

---
**Algorithm 2** IWeS-V

---
**Require:** Labeled Pool $\mathcal{P}$.

1: Initialize: $\mathcal{S}_0 = \emptyset, \mathcal{H}_0 = \mathcal{H}$.
2: **for** $t = 1, 2, \ldots, T$ **do**
3:   Sample $(x_t, y_t)$ uniformly at random from $\mathcal{P}$.
4:   Update:

$$\mathcal{H}_t = \left\{ h \in \mathcal{H}_{t-1} : \frac{1}{t-1} \sum_{i=1}^{t-1} \frac{Q_i}{p_i} \ell(h(x_i), y_i) \leq \min_{h \in \mathcal{H}_{t-1}} \frac{1}{t-1} \sum_{i=1}^{t-1} \frac{Q_i}{p_i} \ell(h(x_i), y_i) + \Delta_{t-1} \right\}$$

where $\Delta_{t-1} = \sqrt{\frac{8 \log(2T(T+1)|\mathcal{H}|^2/\delta)}{t-1}}$.

5:   Set $p_t = \max_{f,g \in \mathcal{H}_t} \ell(f(x_t), y_t) - \ell(g(x_t), y_t)$.
6:   $Q_t \sim \text{Bernoulli}(p_t)$.
7:   **if** $Q_t = 1$ **then**
8:     Set $\mathcal{S}_t = \mathcal{S}_{t-1} \cup \left\{ \left( x_t, y_t, \frac{1}{p_t} \right) \right\}$
9:   **else**
10:     $\mathcal{S}_t = \mathcal{S}_{t-1}$.
11:   **end if**
12:   $h_t = \arg\min_{h \in H} \sum_{(x,y,w) \in \mathcal{S}_t} w \cdot \ell(h(x), y)$.
13: **end for**
14: **return** $h_T$

---

In order to prove Theorem 4.2, we first present the following Lemma.

**Lemma B.1.** Define the weighted empirical loss as $L_{t-1}(f) = \frac{1}{t-1} \left( \sum_{i=1}^{t-1} \frac{Q_i}{p_i} \ell(f(x_i), y_i) \right)$. For all $\delta > 0$ with probability at least $1 - \delta$, for all $t \in \{1, \ldots, T\}$, and for all $f, g \in \mathcal{H}_{t-1}$, we have
$$|L_{t-1}(f) - L_{t-1}(g) - L(f) + L(g)| \leq \Delta_{t-1}.$$

*Proof of Lemma B.1.* Fix any $t \in \{1, \ldots, T\}$. For any $f, g \in \mathcal{H}_{t-1}$, define
$$Z_i = \frac{Q_i}{p_i} \left( \ell(f(x_i), y_i) - \ell(g(x_i), y_i) \right) - (L(f) - L(g))$$
for $i \in \{1, \ldots, t-1\}$. The sequence $Z_i$ is a martingale difference since
$$\mathbb{E}[Z_i | Z_1, \ldots Z_{i-1}] = \mathbb{E}\left[ \frac{Q_i}{p_i} (\ell(f(x_i), y_i) - \ell(g(x_i), y_i) - (L(f) - L(g)) | Z_1, \ldots Z_{i-1} \right] = 0.$$
Due to the fact that
$$|Z_i| \leq \frac{1}{p_i} |\ell(f(x_i), y_i) - \ell(g(x_i), y_i)| + |L(f) - L(g)| \leq 2$$
as $p_i \geq |\ell(f(x_i), y_i) - \ell(g(x_i), y_i)|$ for $f, g \in \mathcal{H}_{t-1}$. Thus, we can apply Azuma's inequality,
$$\Pr[|L_{t-1}(f) - L_{t-1}(g) - L(f) + L(g)| \geq \Delta_{t-1}]$$
$$\leq 2\exp(-(t-1)\Delta_{t-1}^2/8) = \frac{\delta}{T(T+1)|H|^2},$$
where we used the fact that $(x_t, y_t)$ is i.i.d. Since $\mathcal{H}_{t-1}$ is a random subset of $\mathcal{H}$, we take the union bound over $f, g \in \mathcal{H}$ and $t - 1$. Then we take another union bound over $T$ to finish the proof. $\square$

Next we provide the proof of Theorem 4.2.

**Theorem 4.2.** Let $h^* \in \mathcal{H}$ be the minimizer of the expected loss function $h^* = \mathrm{argmin}_{h \in \mathcal{H}} L(h)$. For any $\delta > 0$, with probability at least $1 - \delta$, for any $t \geq 1$ with $t \in \{1, 2 \ldots, T\}$, we have that $h^* \in \mathcal{H}_t$ and that $L(f) - L(g) \leq 2\Delta_{t-1}$ for any $f, g \in \mathcal{H}_t$. In particular, if $h_T$ is the output of IWeS-V, then $L(h_T) - L(h^*) \leq 2\Delta_{T-1} = \mathcal{O}\big(\sqrt{\log(T/\delta)/T}\big)$.

*Proof of Theorem 4.2.* We show that $h^* \in \mathcal{H}_t$ by induction. Base case holds as $h^* \in \mathcal{H}_1 = \mathcal{H}$. Now assuming that $h^* \in \mathcal{H}_{t-1}$ holds, we show that $h^* \in \mathcal{H}_t$. Let $h' = \mathrm{argmin}_{f \in \mathcal{H}_{t-1}} L_{t-1}(f)$. By Lemma B.1,

$$L_{t-1}(h^*) - L_{t-1}(h^{'}) \leq L(h^*) - L(h') + \Delta_{t-1} \leq \Delta_{t-1}$$

since $L(h^*) - L(h') \leq 0$ by definition of $h^*$. Thus, $L_{t-1}(h^*) \leq L_{t-1}(h^{'}) + \Delta_{t-1}$ which means that $h^* \in \mathcal{H}_t$ by definition of $\mathcal{H}_t$.

Since $\mathcal{H}_t \subseteq H_{t-1}$, Lemma B.1 implies that for any $f, g \in \mathcal{H}_t$,

$$
\begin{aligned}
L(f) - L(g) &\leq L_{t-1}(f) - L_{t-1}(g) + \Delta_{t-1} \\
&\leq L_{t-1}(h') + \Delta_{t-1} - L_{t-1}(h') + \Delta_{t-1} \leq 2\Delta_{t-1}.
\end{aligned}
$$

Noting that $h^*, h_t \in \mathcal{H}_t$ completes the proof. $\qquad\square$

## B.3 Proofs for statements in Section 4.2

**Theorem 4.3.** For any $\delta > 0$, with probability at least $1 - \delta$, the expected sampling rate of the IWeS-V algorithm is: $\sum_{t=1}^{T} \mathbb{E}_{(\mathrm{x}_t, y_t) \sim \mathcal{D}} \big[p_t | \mathcal{F}_{t-1}\big] = \mathcal{O}\Big(\theta_{\mathrm{S}} \big(L(h^*)T + \sqrt{T \log(T/\delta)}\big)\Big)$.

*Proof of Theorem 4.3.* By Theorem 4.2, $\mathcal{H}_t \subset \{h \in \mathcal{H} : L(h) \leq L(h^*) + 2\Delta_{t-1}\}$. Using this fact and that $h^*$ is the best in class, it holds that

$$\rho_{\mathrm{S}}(h, h^*) = \mathbb{E}_{(\mathrm{x},y) \in \mathcal{D}} |\ell(h(\mathrm{x}), y) - \ell(h^*(\mathrm{x}), y)| \leq L(h) + L(h^*) \leq 2L(h^*) + 2\Delta_{t-1}.$$

Thus, letting $r = 2L(h^*) + 2\Delta_{t-1}$, it holds that $\mathcal{H}_t \subset \mathcal{B}_{\mathrm{S}}(h^*, r)$. Then,

$$
\begin{aligned}
\mathbb{E}_{(\mathrm{x}_t, y_t) \in \mathcal{D}}[p_t | \mathcal{F}_{t-1}] &= \mathbb{E}_{(\mathrm{x}_t, y_t) \in \mathcal{D}}[\sup_{f,g \in \mathcal{H}_t} |\ell(f(\mathrm{x}_t), y_t) - \ell(g(\mathrm{x}_t), y_t)| | \mathcal{F}_{t-1}] \\
&\leq 2\mathbb{E}_{(\mathrm{x}_t, y_t) \in \mathcal{D}}[\sup_{h \in \mathcal{H}_t} |\ell(h(\mathrm{x}_t), y_t) - \ell(h^*(\mathrm{x}_t), y_t)| | \mathcal{F}_{t-1}] \\
&\leq 2\mathbb{E}_{(\mathrm{x}_t, y_t) \in \mathcal{D}}[\sup_{h \in \mathcal{B}_{\mathrm{S}}(h^*, r)} |\ell(h(\mathrm{x}_t), y_t) - \ell(h^*(\mathrm{x}_t), y_t)|] \\
&\leq 2\theta_{\mathrm{S}} r = 4\theta_{\mathrm{S}}(L(h^*) + \Delta_{t-1})
\end{aligned}
$$

By summing the above over $t \in [T]$, the sample complexity bound of IWeS-V is then given by $\mathcal{O}\Big(\theta_{\mathrm{S}} L(h^*)T + \theta_{\mathrm{S}} \sqrt{T}\Big)$, which completes the proof. $\qquad\square$

**Theorem 4.4.** If the loss function has a bounded slope asymmetry $K_\ell$, then $\theta_{\mathrm{S}} \leq \theta_{\mathrm{AL}} K_\ell$.

*Proof of Theorem 4.4.* We first prove

$$\mathcal{B}_{\mathrm{IWAL}}(h^*, r) \subseteq \mathcal{B}_{\mathrm{S}}(h^*, r) \subseteq \mathcal{B}_{\mathrm{IWAL}}(h^*, K_\ell r). \tag{5}$$

The left hand side follows directly from the definition. For the right hand side, assume $h \in \mathcal{B}_{\mathrm{S}}(h^*, r)$, then by the definition of $K_\ell$, we have

$$r \geq \rho_{\mathrm{S}}(h, h^*) = \mathbb{E}_{(\mathrm{x},y) \in \mathcal{D}} [|\ell(h(\mathrm{x}), y) - \ell(h^*(\mathrm{x}), y)|] \geq \frac{1}{K_\ell} \mathbb{E}_{\mathrm{x} \in \mathcal{X}} \sup_{y \in \mathcal{Y}} |\ell(h(\mathrm{x}), y) - \ell(h^*(\mathrm{x}), y)| = \frac{1}{K_\ell} \rho_{\mathrm{IWAL}}(h, h^*)$$

where the second inequality comes from

$$
\begin{aligned}
K_\ell = \sup_{z,z' \in \mathcal{Z}} & \frac{\max_{y \in \mathcal{Y}} |\ell(z,y) - \ell(z',y)|}{\min_{y \in \mathcal{Y}} |\ell(z,y) - \ell(z',y)|} \\
\geq & \sup_{x \sim \mathcal{X}} \frac{\max_{y \in \mathcal{Y}} |\ell(h(x),y) - \ell(h^*(x),y)|}{\min_{y \in \mathcal{Y}} |\ell(h(x),y) - \ell(h^*(x),y)|} \\
\geq & \mathbb{E}_{x \sim \mathcal{X}} \frac{\max_{y \in \mathcal{Y}} |\ell(h(x),y) - \ell(h^*(x),y)|}{\min_{y \in \mathcal{Y}} |\ell(h(x),y) - \ell(h^*(x),y)|} \\
\geq & \frac{\mathbb{E}_{x \sim \mathcal{X}} \max_{y \in \mathcal{Y}} |\ell(h(x),y) - \ell(h^*(x),y)|}{\mathbb{E}_{x \sim \mathcal{X}} \min_{y \in \mathcal{Y}} |\ell(h(x),y) - \ell(h^*(x),y)|} \\
\geq & \frac{\mathbb{E}_{x \sim \mathcal{X}} \max_{y \in \mathcal{Y}} |\ell(h(x),y) - \ell(h^*(x),y)|}{\mathbb{E}_{(x,y) \sim \mathcal{D}} |\ell(h(x),y) - \ell(h^*(x),y)|}
\end{aligned}
$$

which follows by basic properties of expectations. As a result, we have

$$
\begin{aligned}
\theta_S = \sup_{r \geq 0} & \frac{\mathbb{E}_{(x,y) \sim \mathcal{D}} \left[ \max_{h \in \mathcal{B}_S(h^*,r)} |\ell(h(x),y) - \ell(h^*(x),y)| \right]}{r} && \text{(By definition of } \theta_S) \\
\leq \sup_{r \geq 0} & \frac{\mathbb{E}_{(x,y) \sim \mathcal{D}} \left[ \max_{h \in \mathcal{B}_{\text{IWAL}}(h^*,K_\ell r)} |\ell(h(x),y) - \ell(h^*(x),y)| \right]}{r} && \text{(Use equation (5))} \\
= K_\ell \sup_{r \geq 0} & \frac{\mathbb{E}_{(x,y) \sim \mathcal{D}} \left[ \max_{h \in \mathcal{B}_{\text{IWAL}}(h^*,r)} |\ell(h(x),y) - \ell(h^*(x),y)| \right]}{r} && \text{(Redefine } K_\ell r \text{ as } r) \\
\leq K_\ell \sup_{r \geq 0} & \frac{\mathbb{E}_{x \in \mathcal{X}} \left[ \max_{h \in \mathcal{B}_{\text{IWAL}}(h^*,r)} \sup_{y \in \mathcal{Y}} |\ell(h(x),y) - \ell(h^*(x),y)| \right]}{r} \\
= K_\ell \theta_{\text{IWAL}}. &
\end{aligned}
$$

which concludes the proof. $\qquad\square$

### B.4 THE ENHANCED-IWeS-V ALGORITHM

Here we analyze a modified algorithm, which we called the Enhanced-IWeS-V , that uses new slack term defined as

$$
\Delta_t^{\text{EIWeS}} = \frac{2}{t} \left( \sqrt{\sum_{t=1}^T p_t + 6\sqrt{\log\left((3+t)t^2/\delta\right)}} \right) \cdot \sqrt{\log\left(8T^2|\mathcal{H}|^2 \log(T)/\delta\right)}
$$

to define the version space in the IWeS-V algorithm. The theorem below proves that the Enhanced-IWeS-V admits improved sampling rate bound that is smaller than the bound in Theorem 4.3 since the square root term also scales with $L(h^*)$. In particular, under realizable setting where $L(h^*) = 0$, the sampling rate bound is $\mathcal{O}\left(\log^3(T)\right)$, which depends poly-logarithmic in $T$ and is smaller than the $\mathcal{O}\left(\sqrt{T \log(T)}\right)$ bound from Theorem 4.3.

**Theorem B.2.** For all $\delta > 0$, for all $T \geq 3$, with probably at least $1 - \delta$, the expected sampling rate of the Enhanced-IWeS-V algorithm is:

$$
\sum_{t=1}^T \mathbb{E}_{(x_t,y_t) \sim \mathcal{D}}[p_t | \mathcal{F}_{t-1}] \leq \theta_S \left( L(h^*)T + \mathcal{O}\left(\sqrt{L(h^*)T \log(T/\delta)}\right) \right) + \mathcal{O}\left(\log^3(T/\delta)\right).
$$

*Proof of Theorem B.2.* The proof follows from that of Lemma 6 in Cortes et al. (2019). From Theorem 4.3, for $t \geq 3$,

$$
\sum_{t=1}^T \mathbb{E}_{(x_t,y_t) \sim \mathcal{D}} \left[ p_t | \mathcal{F}_{t-1} \right] = 4\theta_S \left( L(h^*) + \Delta_{t-1} \right).
$$

Plugging in the expression of $\Delta_{t-1}^{\text{ElWeS}}$ into $\Delta_{t-1}$, and applying a concentration inequality to relate $\sum_{t=1}^{T} p_t$ to $\sum_{t=1}^{T} \mathbb{E}_{(\mathsf{x}_t,y_t)\sim\mathcal{D}}\left[p_t|\mathcal{F}_{t-1}\right]$, we end up with a recursion on $\mathbb{E}_{(\mathsf{x}_t,y_t)\sim\mathcal{D}}\left[p_t|\mathcal{F}_{t-1}\right]$:

$$\mathbb{E}_{(\mathsf{x}_t,y_t)\sim\mathcal{D}}\left[p_t\big|\mathcal{F}_{t-1}\right] \leq 4\theta_{\mathsf{S}}L(h^*) + \frac{4\theta_{\mathsf{S}}c_1}{t-1}\sqrt{\sum_{s=1}^{t-1}\mathbb{E}_{(\mathsf{x}_t,y_t)\sim\mathcal{D}}\left[p_s\big|\mathcal{F}_{s-1}\right]} + c_2\left(\frac{\log\left((t-1)|\mathcal{H}|/\delta\right)}{t-1}\right).$$

where $c_1 = 2\sqrt{\log\left(\frac{8T^2|\mathcal{H}|^2\log(T)}{\delta}\right)}$, and $c_2$ is a constant. Then we show for all $t \geq 3$, for constant $c_3 = \mathcal{O}\left(\sqrt{\log\left(T|\mathcal{H}|/\delta\right)}\right)$ and $c_4 = \mathcal{O}\left(\log^2\left(T|\mathcal{H}|/\delta\right)\right)$, we have

$$\mathbb{E}_{(\mathsf{x}_t,y_t)\sim\mathcal{D}}\left[p_t\big|\mathcal{F}_{t-1}\right] \leq 4\theta_{\mathsf{S}}L(h^*) + c_3\sqrt{\frac{L(h^*)}{t-1}} + \frac{c_4}{t-1}$$

The above can be proved by induction as in Cortes et al. (2019). By summing the above over $t \in [T]$ gives us the result. We absorb the dependency on $|\mathcal{H}|$ inside the big-O notation.

$\square$

