# OpenReview forum: "Leveraging Importance Weights in Subset Selection"
_ICLR.cc/2023/Conference — ICLR 2023 poster_

### Official Review · Reviewer_7Jex · 2022-10-24

**Confidence:** 4
**Correctness:** 4
**Technical Novelty And Significance:** 3
**Empirical Novelty And Significance:** 2
**Recommendation:** 8

**Clarity, Quality, Novelty And Reproducibility:**

This paper is well written and clear.  The contributions appear to be novel, with good empirical and theoretical contributions.  Although the authors do not provide source code, enough detail appears to be present in the paper to allow the proposed methods to be reproduced.


**Strength And Weaknesses:**

Strengths:
* IWeS is applicable to arbitrary model families.  It can also be used in an active learning where only unlabeled examples are available.
* IWeS matches or outperforms a number of competing baseline approaches for several datasets on multi-class and multi-label prediction tasks.
* Useful theoretical guarantees, in terms of generalization and sampling rate bounds, are provided for a closely related algorithm, called IWeS-V.

Weaknesses:
* The entropy-disagreement-based variant of IWeS (IWeS-dis) requires training two models, which can be expensive in terms of computation and memory.
* Theoretical guarantees are provided for the IWeS-V algorithm, which is closely related to the IWeS-dis algorithm.  However, theoretical guarantees are not provided for a variant of the IWeS-V algorithm that is closely related to the entropy-based IWeS algorithm (IWeS-ent), which is computationally less expensive since it requires training only one model.
* No detailed analysis of the computational costs associated with the proposed methods is presented in the paper.


**Summary Of The Paper:**

This paper presents an approach for subset selection for training models in a batch setting.  The proposed approach, called IWeS, uses importance sampling to select examples for inclusion in the subset, where the sampling probability is based on the model’s entropy.  Experimental results show that IWeS provides moderate to substantial improvements in predictive performance compared to several competing baseline methods.  Furthermore, a theoretical analysis for a closely related approach, IWeS-V, including generalization and sampling rate bounds.

**Summary Of The Review:**

This is a strong paper, with good empirical and theoretical results.  The experimental results are convincing, showing that the proposed method matches or outperforms a number of baselines.  While there are some issues, as mentioned in the list of weaknesses above, they do not significantly detract from the value of the contributions.  Overall, this is a good paper worthy of acceptance.

---

> ### Author Response · Authors · 2022-11-12
> **Author response**
>
> We thank the reviewer for the helpful feedback and we will address all comments in the final version of the paper.
>
> The primary computational costs of the IWeS-ent algorithm entail, at each sampling iteration r, training the model on previously selected examples and measuring the entropy of an example until k examples are selected. Training the model on previously selected examples to base sampling decisions is a cost that is incurred by all baseline methods that we consider except for random sampling. To measure the entropy of an example, the main computational cost is running model inference. All methods require running inference a constant fraction of examples per round apart from Coreset, which computes a submodular function at least for a constant fraction of examples, and random sampling, which requires no computation. The IWeS-dis algorithm has the same computational costs as IWeS-ent except for an additional training of a secondary model at each sampling iteration r.

---

> > ### Comment · Reviewer_7Jex · 2022-11-17
> > **Rebuttal response**
> >
> > I thank the authors for their rebuttal comments and explanations. I have read all reviews and the rebuttal comments from the authors. I am satisfied with the authors' comments, and my score of 8 (accept, good paper) remains unchanged.

---

### Official Review · Reviewer_q4Qj · 2022-10-25

**Confidence:** 4
**Correctness:** 3
**Technical Novelty And Significance:** 3
**Empirical Novelty And Significance:** 3
**Recommendation:** 6

**Clarity, Quality, Novelty And Reproducibility:**

I think the topic in this paper is interesting and this paper is generally well-written. I have the following comments/questions. I look forward to the response/clarification from the author(s). Thanks.

1. In the introduction, the author(s) mentioned another related method, IWAL. Maybe an empirical comparison with the algorithm IWAL is needed? Or is IWAL difficult to implement in practice?

2. One motivation of subset selection is to find the most informative subset from a large number of training samples to approximate training with the entire training set. So, in experiments, it is necessary to compare the performance on the whole training set, which can be regarded as a baseline to further evaluate the effectiveness of the proposed algorithm.

3. For a better understanding of the theoretical analyses, such as the generalization bound and sampling rate bound, some empirical experiments are needed for further verification.

4. If the input data matrix is transposed, can the work in this paper, i.e., data subset selection, be applied to feature selection?

In addition, the format of the references is quite inconsistent. Please check carefully and correct it.

---




**Strength And Weaknesses:**

### Strength:

1. Selecting an information subset from a large dataset can effectively reduce the computation time and storage costs. In this paper, the author(s) further considered the weights of selected samples.

2. Empirical evaluations are performed for validating the proposed algorithm.

---

### Weaknesses:

The theoretical analysis of the proposed algorithm, such as the generalization bound and sampling rate bound, may require further empirical experiments.

For more details, please see the section of "Clarity, Quality, Novelty And Reproducibility".

---


**Summary Of The Paper:**

This paper focuses on subset selection. Based on general loss functions and hypothesis classes, the author(s) proposed an importance-weighted subset selection algorithm called IWeS. Through Empirical experiments, the author(s) tried to show the advantage of the proposed algorithm.

---


**Summary Of The Review:**

The work in this paper is interesting and it considers weighting the selected samples; However, there are some unclear/inadequate aspects in the description (including experiments) of this paper. Maybe it needs the author(s) to clarify them. Thanks.

In addition, in this paper, the author(s) performed empirical experiments. I am not sure the description would be enough to reproduce since no code seems to be provided (other than some pseudo-code descriptions).

---

---

> ### Author Response · Authors · 2022-11-12
> **Author response**
>
> We will clarify the following aspects brought up by the reviewer:
> 1. IWAL is difficult to implement for deep neural networks since it must maintain a version space, which is computationally infeasible for any class of models except for linear models. Please see the last two sentences in the uncertainty paragraph in the related work section and further discussions in Section 4.
> 2. Although the theoretical guarantees are stated with respect to performance on the full pool,  if we modeled our empirical setting in this fashion, it would result in each sampling algorithm reaching this final performance at a different total number of selected examples.  Our empirical setup reflects a more natural use of the algorithms in a practical setting where all the sampling algorithms have a fixed selection budget. Note that in both formulations, the main insight is that we attain a better model performance per selected examples.
> 3. The IWeS algorithm is a practical implementation of IWeS-V which admits theoretical guarantees. For IWeS, we present plots showing the tradeoff between an empirical estimate of generalization ability (i.e. error on the test set) and number of selected examples.
> 4. Our method could perhaps be modified for feature subset selection, but one main question to answer is how to properly reweight features and what effects that has.
> 5. We unfortunately cannot release the code due to our institution’s policy, but we are happy to further describe any implementation details that are unclear.

---

### Official Review · Reviewer_EWd7 · 2022-11-01

**Confidence:** 3
**Correctness:** 3
**Technical Novelty And Significance:** 2
**Empirical Novelty And Significance:** 3
**Recommendation:** 6

**Clarity, Quality, Novelty And Reproducibility:**

For clarity and quality, the paper is good and clear in presenting the core idea, related work, and baseline methods.

For novelty, the paper's main idea is based on one prior work, but improving the sampling probability to make the method more suitable for deep models. The relationship between prior work and this work is well-explained and discussed in the paper.

For reproducibility, it seems that the authors did not provide the code for reproducing the work, but they presented lots of details for reproducing the proposed method in the paper.

**Strength And Weaknesses:**

Strengths:
1. In this paper, the considered subset selection problem is promising and useful in many cases, such as efficient deep neural network training, limited budget for data storage, and active learning. The proposed method is applicable for deep models, and for both labeled and unlabeled data during the subset selection.
2. The proposed method is theoretically motivated and evaluated by extensive experiments on multi-class datasets and a large-scale multi-label dataset.
3. The paper is well-written and presented. This paper's main messages are clear and easy to follow for readers.

Weaknesses:
1. The core idea of the paper is largely based on prior work (i.e., Beygelzimer et al. (2009)), which may reduce the novelty of this paper. Compared with prior work, this paper contributes to using the model's entropy to define the sampling probability to make the method suitable for deep models.
2. I am wondering if the seed set's size or the quality of pertained models affects the performance of the proposed method. If the seed set is very small, then the model may not be well pretrained (in Line 6, Algorithm 1). In this case, perhaps the data cannot be effectively weighted, especially during the beginning of the training.

Other comments:
In the proposed method, it seems that once one data is selected, its weight would be fixed. However, as the model is trained to become better, the weight of one data may also change if recomputed. Then, for the proposed method, it may happen that the weights for one particular data may differ by the order of the data selection. Would it improves the proposed method if recomputing weights of all existing selected data at some specific training epoch?

**Summary Of The Paper:**

In this paper, the authors proposed a method for subset selection, which is useful for applications such as active learning. Specifically, in the proposed method, when constructing the subset, data is selected by importance sampling, where the sampling probability relates to the model's entropy. In the subset, the selected data is weighted inversely proportional to its sampling probability for computing a weighted loss function. This method is empirically evaluated on six multi-class classification image datasets and a large-scale multi-label Open Images dataset, and shows good performance when compared with baselines. The theoretical motivation of the proposed method is also presented in the paper.

The paper's contributions include a theoretically motivated method for subset selection that is suitable for general losses and hypothesis classes and extensive experimental results for evaluating the proposed method and baselines.

**Summary Of The Review:**

The paper presents a subset selection method suitable for deep models with theoretical motivation and extensive experimental results. Although the paper's main idea is based on one prior work, this paper contributes to using model's entropy to define the sampling probability for the benefit of using deep models. This paper is also well-written and completed.

---

> ### Author Response · Authors · 2022-11-12
> **Author response**
>
> *IWAL vs IWeS:*   The IWeS algorithm is similar to IWAL in that they both do importance sampling, but importance sampling is a mathematical technique that has been around for decades and which has been used in a wide variety of applications (e.g. Monte Carlo methods). The crux and novelty of the IWeS algorithm is determining the best way to both sample and reweight examples for general model classes such as deep neural networks. IWAL is more similar to IWeS-V except that IWeS-V leverages the label information available at sampling time. As the number of classes grows, the sampling probabilities of IWAL and IWeS can differ significantly.
>
> *Seed set size:*  If the seed set size is too small, it could affect IWeS and other sampling algorithms, but empirically, we find that if we set seed set size to be at least on average 50 examples per class for the multi-class datasets, then the algorithm performs well.
>
> *Recomputing weights:* To have an unbiased estimator, the weights need to be 1 over the sampling probability of the example and thus by recomputing weights a posteriori, we would lose the unbiasedness of the estimator. Nevertheless, from a practical standpoint it would be interesting to see the effects of recomputing weights on model performance.

---

> > ### Comment · Reviewer_EWd7 · 2022-11-22
> > **Post-rebuttal comments**
> >
> > Thank the authors for the rebuttal! After reading the rebuttal and comments from other reviewers, I maintain my initial evaluation. I would not strongly vote for an acceptance of this work.
> >
> > My biggest concern about this paper is still the novelty issue. As I mentioned before, the paper is based on Beygelzimer et al. (2009), but contributes to using the model's entropy to define the sampling probability for the ease of using deep neural networks. However, using entropy-based uncertainty sampling is quite a straightforward idea in active learning. Applying this technique to the existing work may not be a big contribution to the community. I agree with Reviewer j2Bp on this point.
> >
> > The advantage of the paper is that it is well-written. The main messages are clearly conveyed in this paper. Also, the paper includes empirical evaluations of the proposed method on many benchmark datasets. The baselines are selected from both subset selection and active learning literature and may not be the latest.
> >
> > Reference: Beygelzimer et al. Importance weighted active learning. In ICML 2009.

---

### Official Review · Reviewer_j2Bp · 2022-11-02

**Confidence:** 4
**Correctness:** 2
**Technical Novelty And Significance:** 2
**Empirical Novelty And Significance:** 2
**Recommendation:** 3

**Clarity, Quality, Novelty And Reproducibility:**

The presentation of this paper is clear.

However, the contribution of this paper is insufficient for the following reasons: 1) The paper does not propose new techniques, e.g., novel sampling strategies. The main idea of the proposed two strategies is similar to the previous works. 2) The main theoretical results are applications of the prior works.

**Strength And Weaknesses:**

Strength

The paper proposes the entropy-based disagreement sampling strategy, which is slightly different from the previous sampling strategies.

The paper conducts comparison experiments on a large-scale dataset OpenImage, which is relatively realistic.

Weakness

The main idea of the proposed two strategies is similar to the previous strategies, such as uncertainty sampling and largest margin sampling [1]. It seems that the paper does not propose some new techniques. I think its technical contribution is limited.

[1] Active learning literature survey

The theoretical analyses proposed in the paper are also trivial. The main results follow the previous works [1][2]. Furthermore, the main theoretical results are based on the unbiased estimator, which has been first proposed in [2].


[1] Active Learning for Convolutional Neural Networks: A Core-Set Approach
[2] Importance Weighted Active Learning

The experiments are weak in some extent due to the lack of comparison with sota methods. In fact, besides two baselines uncertainty sampling and random sampling, the proposed method is only compared with two methods, BADGE (2019) and Coreset (2017). These methods have been proposed three years ago. The paper is suggested to compare with more recent methods [1]

[1] Influence selection for active learning, 2021
[2] Task-aware variational adversarial active learning, 2021


**Summary Of The Paper:**

The paper proposes an active learning method by utilizing importance reweighting technique. Authors derive two strategies, including entropy-based disagreement and entropy to sample examples. Then, authors reweight losses by the sampling probability of each example to achieve an unbiased estimator. Experiments are conducted on multiple benchmark datasets.

**Summary Of The Review:**

The technical contribution is insufficient and the novelty is limited.
Although the paper reports the extensive experimental results, it suffers from the lack of comparisons with some sota methods.

---

> ### Author Response · Authors · 2022-11-12
> **Author Response**
>
> We thank the reviewer for their detailed comments and clarify several points below.
>
> *Uncertainty sampling vs IWeS:* Uncertainty sampling does not reweight selected examples, which is a key component of IWeS. Additionally, unlike IWeS, uncertainty sampling or variants thereof do not admit theoretical guarantees for general data distributions and general loss functions. In fact, deriving theoretical guarantees for uncertainty sampling in the agnostic setting has been a long standing open question.
>
> *IWAL vs IWeS:*  The IWeS algorithm is similar to IWAL in that they both do importance sampling, but importance sampling is a mathematical technique that has been around for decades and which has been used in a wide variety of applications (e.g. Monte Carlo methods). The crux and novelty of the IWeS algorithm is determining the best way to both sample and reweight examples for general model classes such as deep neural networks.
>
> *Theory:* None of our theoretical results follow from the Coreset paper and the unbiased estimator is a direct consequence of importance sampling, i.e. it was not first introduced in the IWAL paper. The novelty of our theoretical results lies in correctly leveraging the label information which is not available in an active learning setting. In doing so, we not only obtain better sampling rate results, but also introduce a new disagreement coefficient for subset selection which can be of independent interest to the community.
>
> *Additional baselines:*  We thank the reviewer for suggesting additional relevant baselines. As discussed in the related work section, we pick baselines that are representative methods falling into three categories: uncertainty, diversity, loss gradient. The papers [1,2] suggested by the reviewer also fall into these categories and we will add them to the related work section.
>
> [1] belongs to the loss gradient category since they leverage the gradient / Hessian information.
> [2] is a combination of [3] and [4], belonging to uncertainty category with a focus on imbalanced datasets.
>
> BADGE and Coreset are considered strong baselines and below, [5,6,7,8,9] are recent papers that appeared in 2022 ICLR/ICML/NeurIPS that use BADGE and Coreset as main performance comparators.  Note that these recent papers do not compare with the methods in [1,2].
>
> [1] Liu, Zhuoming, et al. "Influence selection for active learning." Proceedings of the IEEE/CVF International Conference on Computer Vision. 2021.
>
> [2] Kim, Kwanyoung, et al. "Task-aware variational adversarial active learning." Proceedings of the IEEE/CVF Conference on Computer Vision and Pattern Recognition. 2021.
>
> [3] Yoo, Donggeun, and In So Kweon. "Learning loss for active learning." Proceedings of the IEEE/CVF conference on computer vision and pattern recognition. 2019.
>
> [4] Sinha, Samarth, Sayna Ebrahimi, and Trevor Darrell. "Variational adversarial active learning." Proceedings of the IEEE/CVF International Conference on Computer Vision. 2019.
>
> [5] Wang, Haonan, et al. "Deep Active Learning by Leveraging Training Dynamics." Advances in Neural Information Processing Systems (2022).
>
> [6] Mohamadi, Mohamad Amin, Wonho Bae, and Danica J. Sutherland. "Making Look-Ahead Active Learning Strategies Feasible with Neural Tangent Kernels." Advances in Neural Information Processing Systems (2022).
>
> [7] Elenter, Juan, Navid NaderiAlizadeh, and Alejandro Ribeiro. "A Lagrangian Duality Approach to Active Learning." Advances in Neural Information Processing Systems (2022).
>
> [8] de Mathelin, A., Deheeger, F., Mougeot, M., & Vayatis, N. Discrepancy-based active learning for domain adaptation. In International Conference on Learning Representations (2021).
>
> [9] Hacohen, Guy, Avihu Dekel, and Daphna Weinshall. "Active learning on a budget: Opposite strategies suit high and low budgets." arXiv preprint arXiv:2202.02794 (2022).

---

### Official Review · Reviewer_DSZY · 2022-11-26

**Confidence:** 3
**Correctness:** 4
**Technical Novelty And Significance:** 3
**Empirical Novelty And Significance:** 3
**Recommendation:** 8

**Clarity, Quality, Novelty And Reproducibility:**

The paper is written well in general. However, I have a minor concern about the presentation of the contribution. The method is presented as a subset selection method of the training sample. But, then, the standard stochastic gradient methods are in the same category and I wonder if those needs to be compared as well. As with other previous work, I feel more convinced if the proposed method is presented as an active learning method for querying labels.

I think the methods are novel but not significantly innovative (motivated by several previous results). The experimental section seems to contain enough information to reproduce the results.

**Strength And Weaknesses:**

Strength:
The proposed method shows remarkable improvements over previous work in practice. Although the method does not have any theoretical guarantees, but it is motivated by the other theoretically guaranteed algorithm.

Weaknesses:
The weakness of the paper is, as mentioned above, the lack of theoretical guarantees. However, other previous work does not have guarantees either. So, this is not a disadvantage over previous work.



**Summary Of The Paper:**

The paper proposes generic active learning methods for any learning method.  The main method is motivated by the other method with theoretical guarantees. More precisely, the main proposed method is a practical heuristic one. The proposed method is compared with other previous active learning methods and naive ones and shows better performances on the benchmark data sets.

**Summary Of The Review:**

The paper proposes a practical heuristic method for active learning by querying labels. The paper shows non-trivial practical improvements over previous work.

---

### Decision · Program_Chairs · 2023-01-20

**Decision:**

Accept: poster

**Justification For Why Not Higher Score:**

Thanks for the detailed replies to the reviewers, which clarified several issues raised in the initial reviews. Overall the reviewers satisfied with the author responses and thus I recommend the acceptance of this paper. However, the authors' implementation is not clear from the paper and thus I explicitly request the authors to make their experiment code public so that readers can reproduce the results provided in the paper.

**Justification For Why Not Lower Score:**

Even though there is no theoretical guarantee, the main idea is interesting and a good contribution to the active learning community.

**Metareview: Summary, Strengths And Weaknesses:**

Summary:
The paper proposes generic active learning methods for any learning method.

Strength:
The proposed algorithm does not require maintaining a version space, which has an advantage for general hypothesis classes, and it works well in experiments.

Weakness:
The proposed method lacks theoretical guarantees and technical novelty is rather limited.


**Note From Pc:**

if the above contains the word "oral" or "spotlight" please see: "oral" presentation means -> notable-top-5% and "spotlight" means -> notable-top-25%. As stated in our emails, we are disassociating presentation type from AC recommendations

---

> ### Author Response · Authors · 2023-03-01
> **Regarding release the code**
>
> We, unfortunately, cannot release the code due to the institution’s policy. We tried our best to provide enough implementation information in the final version, and we are happy to further describe any unclear details.